# Long-Term Fish Oil Supplementation Attenuates Spike Wave Discharges in the Amygdala of Adult Rats with Early-Life Febrile Seizures

**DOI:** 10.3390/brainsci15040395

**Published:** 2025-04-14

**Authors:** Leopoldo Eduardo Flores-Mancilla, Marisela Hernández-González, Miguel Ángel Guevara-Pérez, Herlinda Bonilla-Jaime, Noemí Gaytán-Pacheco, Claudia Araceli Reyes-Estrada, Fermín Paul Pacheco-Moisés

**Affiliations:** 1Unidad Académica de Medicina Humana y Ciencias de la Salud, Universidad Autónoma de Zacatecas, Ejido la Escondida, Zacatecas 98160, CP, Mexico; 2Instituto de Neurociencias, Centro Universitario de Ciencias Biológicas y Agropecuarias, Universidad de Guadalajara, Francisco de Quevedo No. 180, Col. Arcos Vallarta, Guadalajara 44130, CP, Mexico; maricela.hgonzalez@academicos.udg.mx; 3Departamento de Biología de la Reproducción, División Ciencias Biológicas y de la Salud, Universidad Autónoma Metropolitana, av. San Rafael Atlixco No. 186, Col. Vicentina, Alcaldía Iztapalapa, Ciudad de México 09340, CP, Mexico; maguevara1954@gmail.com (M.Á.G.-P.); bjh@xanum.uam.mx (H.B.-J.); 4Unidad Académica de Ciencias Químicas, Universidad Autónoma de Zacatecas, Ejido la Escondida, Zacatecas 98160, CP, Mexico; noemigaytan@uaz.edu.mx (N.G.-P.); c_reyes13@uaz.edu.mx (C.A.R.-E.); 5Departamento de Química, Centro Universitario de Ciencias Exactas e Ingenierías, Universidad de Guadalajara, Guadalajara 44430, CP, Mexico; fermin.pacheco@academicos.udg.mx

**Keywords:** febrile seizures, amygdala, θ-3, polyunsaturated fatty acids, electroencephalogram, neuroprotection, absence epilepsy

## Abstract

*Background and Objectives:* Febrile seizures (FS) are neuronal disturbances frequently associated with abnormal electroencephalographic activity (EEG) as spike-wave discharges (SWDs). Fish oil (FO) has high amounts of omega-3 fatty acids (θ-3), and its effects on FS alterations are poorly understood. The aim of this work was to evaluate the effect of long-term FO supplementation on the EEG of the amygdala of adult male rats with early-life FS. *Materials and Methods:* Progenitor female Wistar rats, from puberty to gestation and delivery, were fed daily with a commercial diet supplemented with either fish oil (FO), palm oil (PO), or deionized water (CTRL). After parturition, male pups were exposed for 30 min to hyperthermia (HP) and then returned to their dams. After weaning, pups were fed a commercial diet and the respective treatments up to 155 days of age when electrodes were implanted in the amygdala. *Results:* During early life HP, the PO and CTRL groups reached maximal core temperature (CT) in comparison with the FO group. Furthermore, the FO group only has fewer myoclonus and long latency to adopt an uncontrolled posture. At an adult age, the FO group with early-life FS scored shorter periods of SWDs in amygdala EEG but without seizures and presented minor values of absolute power than the PO and CTRL groups. *Conclusions:* In adult rats, the long-term supplementation of FO minimizes the deleterious behavioral effects caused by early-life FS and decreases the occurrence and amplitude of SWDs in the EEG of the amygdala.

## 1. Introduction

Febrile seizures (FS) are common prevalent seizures that occur in children under the age of six years [1], with a 2–5% incidence in Western countries [2]. FS is defined as seizures accompanied by a fever of at least 100.4 °F (38 °C) with no infection of the Central Nervous System (CNS) [3]. Though usually considered benign, an increased risk of subsequent non-FS occurs in ~40% of the affected population [4,5]. The pathogenesis and neural mechanisms that underlie FS are still under study. Still, several structural and molecular alterations of brain functions and neuronal tissue following FS have been identified, including altered neurogenesis and neuronal sprouting [6] and GABA-A receptor mutations [7], among others.

Early alterations in the brain of rats have significant deleterious consequences caused by exacerbated abnormal neuronal connectivity [8]. It has been suggested that aberrant sprouting might be the basis of abnormal neuronal circuitry that generates posterior epileptogenesis [9]. Several studies have reported the relation between early-life FS and the incidence of epileptic syndrome and subsequent temporal lobe epilepsy (TLE) [10,11]. FS can critically alter the balance of excitation and inhibition on brain structures, possibly leading to hyperexcitability with abnormal EEG [12,13]. For example, children with uncontrolled epilepsy and early histories of FS presented exacerbated abnormal theta activity and elevated responses to nociceptive stimuli while asleep. In that case, EEG abnormalities were observed in 232 (62%) of 373 children with FS [14]. In another study, spike-wave discharges (SWDs) were observed in the EEG of patients [15] and also in an animal model with SWD traces on EEG [16].

SWDs are defined as rhythmic oscillations composed of spikes followed by slow waves generated by thalamocortical circuitry and represent synchronous neural ensemble activity that lasts for several seconds (regularly up to 10), associated with immobility and unresponsiveness. The SWDs are present in two rat strains: GAERS (Genetic Absence Epilepsy Rat from Strasbourg [17]) and Wag/Rij (Wistar Albino Glaxo Rats from Rijswijk [18]). Because the rats of both strains exhibit spontaneous SWDs on EEG but without seizure behavior, they have been used as genetic animal models of absence epilepsy. These animals display generalized SWDs at 7–12 Hz during periods of immobility called behavioral arrest—a pause in activity marked by a frozen posture and a condition that may occur with eyes closed or open [19]. This EEG abnormality has been observed in patients with absence epilepsy, where spontaneous SWDs occur synchronously in both hemispheres in association with a temporary loss or ‘absence’ of consciousness [20]. Studies in rats have reported the arrest of locomotion concurrently with paroxysmal EEG theta bursts but before generalized seizures [21].

Several paradigms have been developed in animal models to study the FS between them and the induction of FS by mean hyperthermia (HP) [22] during early life. This model is based on the fact that, for example, in newborn rats, the key enzyme glutamate decarboxylase (GAD), which catabolizes the glutamate neurotransmitter to GABA, possesses high thermolability, so that the decrease of GABA in the brain, provoke the hyperexcitability characteristic of the FS. Usually, models of HP models are performed in pups after ten days of age [23,24]. By extrapolation, in terms of findings for human beings, the effect of HP at five days of age in rats has also been used [25] and the effects on GAD inhibition by body temperature [26]. It has also been suggested that seizure progression appears to emerge from the limbic circuit after 8–10 months of early-life FS and is induced by an airstream in rats [22].

Currently, it is difficult to prevent recurrent FS because daily adverse therapy antipyretics or drugs such as benzodiazepines have been found to be ineffective or unjustified [27], so preventive therapies are necessary. Natural substances like polyunsaturated fatty acids (PUFAs) θ-3 have been explored with promising results on several alterations of the CNS [28]. PUFAs like docosahexaenoic acid (DHA) and eicosapentaenoic acid (EPA) have been studied for their efficacy in controlling the frequency of seizures in epileptic patients [29,30], and on the threshold and severity of seizures in animal models of epilepsy [31,32]. DHA and EPA are not synthetized de novo inside the body, so they must be consumed in the daily diet [33]. DHA intake during pregnancy in female mice can increase the levels of DHA and estrogens in the cerebral cortex and decrease the susceptibility to FS in offspring [34]. Similarly, one study reported the suppressive effects of DHA intake on increased seizure susceptibility in the PTZ mouse model after growth due to FS [35]. In addition, one study found that daily supplementation (1200 mg) of FO in children with intractable epilepsy decreased seizure periods during three months of treatment [36].

On the other hand, a minimum of four months of FO supplementation was necessary to raise the amygdaloid seizure threshold in rats classified as “resistant” to kindling seizures [37]. Also, FO diminishes the formation of lesions in the brain caused by pilocarpine [38]. Our team reported that a minimum of six months of FO supplementation was required to attenuate seizures induced by convulsant 3-mercaptopropionic acid on behavior and amygdala EEG [39].

Considering that abnormal brain functionality as a consequence of early-life FS has been consistently reported and that long-term supplementation with PUFAs has beneficial effects on early-life cerebral insults and posterior damage on neuronal function, the aim of the present study was to evaluate the effect of long-term FO supplementation (from the stage of fetal development through the mother) on the severity of FS induced by early-life HP in male rat pups and EEG activity in the amygdala during spontaneous arrest behavior during adulthood.

## 2. Materials and Methods

### 2.1. Nutritional Supplements

Fish oil (FO) was purchased from Omega/Rx^®^ Zone Labs., Peabody, MA, USA. It is a marine concentrate, purified oil made from anchovies and sardines mixed with tocopherols, ascorbyl palmitate, and citric acid (a 5-mL ration provides 1.8 g of EPA and 0.9 g of DHA). Palm oil (PO) was purchased from Spectrum Chemical Mgf. Corp. (Gardena, CA, USA, but does not contain omega-3. Purina^®^ rat chow was purchased from Purina Mills Co. (Gray Summit, MO, USA. It contains 234.0 g of protein per kg of diet, 45.0 of fat, 623.5 of carbohydrates, 58.0 of fiber, 0.3 of vitamins, and 39.2 of minerals.

### 2.2. Subjects and Treatments

All rats were handled following the protocols for animal care established by the local ethics committee in accordance with NIH specifications and Mexican laws on the care of laboratory animals (NOM-062-ZOO-1999, SAGARPA). The rats were used only once to minimize suffering. The doses of PO and FO (300 mg/kg) were determined according to the dosages and effects reported in a previous study [39,40]. Water and Purina^®^ rat chow were available ad libitum to all groups.

Thirty female Wistar rats were housed under controlled environmental conditions (0700–1900 h light/dark cycle, room temperature 23–24 °C). At 45 days of age, they were separated randomly to form three groups (*n* = 10 per group) and placed in polycarbonate cages (29 × 10 × 10 cm). The treatments assigned included the following: the CTRL group was fed with a normal chow diet complemented by the daily administration of a 300 µL vehicle solution (distilled water); the PO group received a normal chow diet supplemented with palm oil (300 mg/kg); and the FO group was also fed the normal chow diet and fish oil supplement (300 mg/kg). The treatments were administered daily by oral gavage between 21:00–22:00 h. Body weight and food consumption were recorded daily up to 90 days of age when the females of all three groups were allowed to mate with adult male Wistar rats. Future dams will continue to receive the same dietary regimen throughout gestation and nursing. The day of parturition was taken as day 0 for the newborn pups. The litters from each group were mixed and culled to 10 animals. Dams and pups were held in a room under controlled environmental conditions. On the day of the experiment, 10 male pups (five days old) from each group (CTRL, PO, FO) were chosen at random and assigned to the group exposed to HP (at five days of age). Similarly, 10 male pups from each group were selected and assigned to the groups that were not exposed to HP. Induction of HP was performed following the method described in [41] with minimal adaptations.

### 2.3. Induction of Hyperthermia

The pups were placed at the bottom of a glass container (3 L) and exposed to an airstream (from a standard hair dryer) blown into the top of the container (50 cm above the pups). The intensity and temperature of the airstream were programmed to gradually increase the core temperature of the pups. Temperature was monitored using a digital thermometer to ensure it did not exceed 45 °C. The core temperature (CT) of each pup was verified at 2 min intervals with a rectal thermometer (Thermocouple Thermometer 51/52 II, Fluke Corp. (Everett, WA, USA). Values were recorded from onset to the end of the experiment. The HP condition was maintained for 30 min, the period required to gradually raise the rats’ body temperature to 41–42 °C. The pups’ behavior during HP was monitored continually by the experimenters to prevent excessive suffering or death. At the beginning of each experiment, the glass container was rinsed and dried so the temperature of the bottom remained at ~20 °C. The behavioral parameters recorded in the HP groups were myoclonus (intermittent contraction of body members), sudden arrest (when a pup displayed periods of immobility and paused in its activity), and uncontrolled posture (when they were unable to coordinate the march and fell to the floor following myoclonus).

All experiments were videotaped for posterior analysis by a blind, neutral observer. At the end of the period of HP exposure, each pup was moved to a cool surface until its CT normalized to a range of 32–33 °C. At that point, they were carefully rehydrated with water orally. The index of mortality after exposure to HP was low (3%). Two subjects died in the CTRL and PO groups, respectively, but only one in the FO group.

The parallel experiment with the FO, PO, and CTRL pups that were not exposed to HP (NT groups) was performed under similar conditions, as they were also placed in the bottom of the glass container but were not exposed to hot air flow from the hair dryer. At the end of the experiment, both the exposed and unexposed pups were returned to their respective dams until the day of weaning (day 26), when they were separated from their dams and housed in cages under the same conditions of room, diet, and supplementation as their mother until they reached 150 days of age. On the date the rats of the exposed and unexposed groups reached a body weight of 350–400 g, electrodes were implanted bilaterally into their basolateral amygdala. Daily food consumption and body weight continued to be recorded.

### 2.4. Surgical Implantation of the Electrodes

At 150 days of age, the adult rats from the exposed and unexposed groups were taken to an operating room, anesthetized with an i.p. injection of sodium pentobarbital 35 mg/kg) (Anestesal, Pfizer^®^, New York, NY, USA) and maintained at a surgical level of anesthesia for the subsequent intramuscular injections of a tiletamine/zolazepam mixture (Zoletil, Virbac^®^, Carros, France). Monopolar electrodes were implanted permanently simultaneously in both hemispheres, specifically, in the right and left basolateral amygdala (2.8–3.3 mm posterior to the bregma, 4.8 mm lateral to midline, and 8.5 mm below the dura mater) with incisor bars set at –3.3 mm, according to Paxinos and Watson stereotaxic atlas [42]. Stainless steel wire electrodes (diameter = 0.2 mm) were insulated with epoxy resin, except for a small recording area on the tip. This type of monopolar electrode facilitates obtaining EEG recordings over several days without causing interference. A reference electrode was implanted on the nasal bone, and a ground electrode on the occipital bone. All implants were soldered onto connecting pins and fixed to the surface of the skull with acrylic cement. The animals received penicillin (60,000 U, i.m.) to prevent post-implantation infection and were allowed 8–10 days to recover before testing began.

### 2.5. EEG Recording

On the day prior to the experimental session (day 1), the rats from the three groups that suffered HP and the three NT groups were allowed to adapt to the experimental room (22–23 °C) while connected to the polygraph for 1 h (1100–1200 h) inside a sound attenuated, electrically shielded cage (60 × 40 × 30 cm). For EEG recording, the cable connecting the electrodes was coupled to a slip-ring so the rats could move freely over the floor of a clear Plexiglass cage. The cable was connected to the AC preamplifiers of a Grass 7B polygraph (band pass 3–30 Hz). Outputs were plugged into a PCL-812 analog-to-digital converter (Advantech, Co. Taipei, Taiwan) that operated as the interface with a microcomputer. All EEG signals were recorded at a sampling rate of 256 Hz, calibrated with a pulse of 50 microvolts (μV) produced by the preamplifiers, and delivered to a PC as a reference to convert the output from the converter into μV. Capture was performed using a board unit with eight on/off buttons connected to the digital input lines of the converter, configured to capture 2 s segments of EEG that corresponded to episodes of sudden arrest. EEG recording began when a certain button on the board was pressed. A different button was pressed to end the signal input. The computer program Captusen [43] allowed us to simultaneously perform bilateral recording of the left and right amygdala EEG signals while maintaining precise temporal relations with the behavioral conditions. Several EEG segments were captured and stored independently in files for offline analysis. The rats’ behavior was videotaped for posterior analysis.

### 2.6. Behavioral Evaluation During EEG Recording

Each rat was introduced into the chamber and videotaped under conditions of free behavior, understood as when it ran around and showed alertness to external stimuli. Sudden arrest was judged to occur when the rat paused its motor activity and adopted a frozen posture with eyes closed or open [16]. Another study called this condition absence-like seizures [44]. The evaluation period was 30 minutes, which was sufficient to assess each rat’s behavior, while EEG from the left and right amygdalas were recorded. All sessions were videotaped for later analysis by a neutral observer.

### 2.7. EEG Capture and Analysis

EEG recordings of the adult rats with or without an early history of HP were captured during the sudden arrest condition. All EEG traces were inspected carefully before analysis to discard any that contained artifacts. At least 10 2 s EEG segments (max. 20 s) from each rat and condition were included (in the case of the HP groups: before, during, and after the SWDs). Spectral frequencies of 4–25 Hz were calculated by Fast Fourier transformation (FFT). AP (defined as the power density of each frequency band expressed in microvolts squared, μV2/Hz) was obtained for each Hz value using the EEG-Magic computer program [45].

### 2.8. Histology

After EEG recording, the rats were deeply anesthetized with sodium pentobarbital. An intracardial infusion of isotonic saline (0.9%), followed by a 5.0% buffered paraformaldehyde solution, was infused to fix the brain, which was subsequently removed and stored in formalin for at least 2 weeks. At that time, 50-micron-thick sections were sliced with a microtome and stained with Cresyl violet. Inspection under a stereoscopic microscope to trace the stereotaxic coordinates made it possible to reconstruct the path marked by the recording electrode. Only the recordings obtained from the left and right basolateral amygdala, or close to its boundaries, were included in the data analysis.

### 2.9. Statistical Analysis

For the three HP and three NT groups, a two-way test for CT and a one-way ANOVA for the behavioral parameters (sudden arrest, myoclonus, posture loss) were performed, respectively. A Tukey’s test was used for post-hoc comparisons between the pairs of means. Significance was set at *p* < 0.05. EEG segments from each rat in the HP and NT subgroups were inspected by a blind observer. The segments were limited to a range of 4–25 Hz so that no noise produced by cable movements could affect the recordings and subjected to statistical analysis. A separate one-way ANOVA was used to compare the AP of each Hz (from 4–25 Hz) to detect differences among the HP subgroups. A similar one-way ANOVA was applied to the data of the NT subgroups under the arrest condition. Analyses used only EEG segments from before, during, and after SWD events. A Tukey’s test was used for post-hoc comparisons between pairs of means with significance set at *p* < 0.05.

## 3. Results

### 3.1. Effects of Exposure to Hyperthermia on Core Temperature

No statistical differences in CT values were observed in the FO, PO, and CTRL groups at the onset of HP exposure (F(2,45) = 0.74, *p* = 514.87), but with time, the CT of the rats in all groups increased gradually, up to 40–42 °C. The FO group, however, registered values of just 38–39.5°C from min 18 to the end of the experiment (30 min). Significant differences with respect to the CTRL (41.11 ± 0.26 °C) and PO (41.52 ± 0.26 °C) groups were observed (F(2,45) = 6.85, *p* < 0.01) (HSD = 3.45) (Figure 1A).

### 3.2. Effects of Exposure to Hyperthermia on the Pups’ Behavior

At the onset of HP exposure, all pups displayed normal coordination of movements, but as their CT increased gradually, sudden arrest was observed in all groups (Figure 1B). The maximal numbers of sudden arrest events occurred in the CTRL group (28.6 ± 1.6) compared to the PO (17.6 ± 0.82) and FO (16.8 ± 0.9) groups (<0.05) (Figure 1B). In addition, a significantly low number of myoclonus events was observed in the FO group (41.9 ± 2.9) compared to the PO (41.9 ± 2.9) and CTRL (71.4 ± 3.8) groups (Figure 1C).

The effects of HP exposure on the coordination of movements and posture at the onset of the experiment showed that all pups showed normal coordination of walking, but when their CT increased, the inability to coordinate or uncontrolled posture was observed in all groups. However, the pups of the FO group registered a longer latency (min) before presenting uncontrolled posture (18.9 ± 1.3) than PO (7.5 ± 3.1) and CTRL (10.9 ± 3.2) (*p* < 0.05) (Figure 2A). The FO group also had fewer events (11.6 ± 3.2) of uncontrolled posture than the PO (33.8 ± 4.2) and CTRL (29.6 ± 3.6) (*p* < 0.05) groups (Figure 2B), as well as a lower duration of uncontrolled body posture (12.7 ± 1.7) than the PO (24.5 ± 1.6) and CTRL (20.9 ± 2.1) groups (F(2,45) = 48.80, *p* < 0.01 (Figure 2C).

In the CTRL, PO, and FO rat groups that did not suffer HP (NT), no significant differences were observed in the CT or the behavior parameters recorded.

### 3.3. Histological Verification of the Electrode Tips

For the EEG analysis, only the segments obtained from the electrode tips placed in the left and right basolateral amygdala of each rat were considered. Figure 3 shows the trajectory of the electrodes and the location of their tips (Figure 3).

### 3.4. Body Weight and Food and Water Intake of the Groups with and Without a History of HP up to Adulthood

No significant differences were observed among the groups in terms of the initial or final body weight during the growth and maturation of the NT and HP groups. At the end of the study period, the body weight was similar (CTRL: 380 ± 13; PO: 392 ± 10; FO: 385 ± 9 g), as was food and water intake in all groups. No significant differences were obtained.

### 3.5. Behavioral Evaluation of the Normothermic Groups

No significant differences in mobility or the number of sudden arrest events were observed in the adult NT groups of rats; specifically, during the evaluation of free behavior recorded by the camera, all values were similar for the CTRL, PO, and FO groups. No features of spontaneous seizures were observed.

### 3.6. Evaluation of the EEG from the Amygdala of the Normothermic Groups

A visual evaluation of the EEG of the amygdala of the adult rats without a history of HP did not detect any abnormalities in the EEG traces from either the right or left amygdala during sudden arrest behaviors (Figure 4).

Fourier analysis of the EEG from the amygdala of the rats in the three NT groups without HP did not show any differences in the AP of the frequencies from 4 Hz [F(2,12) = 1.106, *p* = 0.91811] to 25 Hz [F(2,12) = 0.412, *p* = 0.69128] in the right and left amygdala (data not graphed).

### 3.7. Behavioral Evaluation of the Adult Rats with Early-Life History of HP

The analysis of the EEG from the amygdala of the rats with a history of HP during arrest behavior revealed abnormalities that were consistent with SWDs. They were observed in seven animals in the CTRL group (70%) and six in the PO (80%) group, but only five in the FO (50%) group. SWDs were observed spontaneously during sudden arrest behaviors in both the right (Figure 5A) and left (Figure 5B) hemispheres. Interestingly, a synchronization of the EEG of the amygdala of both hemispheres was observed during traces but without the characteristic manifestations of seizures.

Analysis of each SWD displayed in the right (Figure 6A) and left amygdala (Figure 6B) showed that PO presented a significant increase of such events during the evaluation period (Figure 6C), a short latency to the onset of SWDs (Figure 6D), and more total events (Figure 6E) than FO and CTRL (F(2,27) = 30.2, *p* < 0.01). The post-hoc analysis showed that the animals in the FO group had a greater latency to the onset of SWDs (Figure 6C) and fewer such events (Figure 6D) than PO and CTRL (HSD = 129.50), *p* < 0.05).

### 3.8. Analysis of the EEG of Amygdala Prior to SWDs in Adult Rats with Early-Life History of HP

EEG analysis before the SWD showed no significant differences in AP in the right amygdala from 4 [F(2,12) = 1.442, *p* = 0.18732] to 25 Hz [F(2,12) = 0.451, *p* = 0.66406]. However, significant differences were detected in the left amygdala, as FO showed a lower AP than PO and CTRL from 4 Hz [F(2,12) = 3.011, *p* < 0.01)] to 12 Hz [F(2,12) = 2.933 *p* < 0.01]. No differences between PO and CTRL were observed [F(2,12) = 1.106, *p* = 0.91811] (Figure 7).

### 3.9. Analysis of the EEG from the Amygdala During SWDs in Rats with Early-Life History of HP

Significant differences in AP in the right and left amygdala during the SWDs were observed from 4 [F(2,12) = 5.85, *p* < 0.1] to 25 Hz [F(2,12) = 6.15, *p* < 0.1]. The post-hoc test showed that FO registered lower AP values than PO and CTRL from 4 Hz [F(2,12) = 4.137, *p* < 0.01] to 25 Hz [F(2,12) = 3.817 *p* < 0.01], though no differences were observed between PO and CTRL from 4 Hz [F(2,12) = 0.888, *p* = 0.78032] to 25 Hz [F(2,12) = 0.775, *p* = 0.46078] (Figure 8).

### 3.10. Analysis of the EEG from the Amygdala After the SWDs in Rats with Early-Life History of HP

No significant differences among the groups were obtained for AP in the EEG from the right amygdala from 4 Hz [(F(2,12) = 1.190, *p* = 0.26822) to 25 Hz [(F(2,12) = 0.071, *p* = 0.94537]; however, in the left amygdala of the FO group, lower AP values were observed than in the PO and CTRL groups from 4 Hz [(F(2,12) = 3.04, *p* < 0.01] to 12 Hz [(F(2,12) = 2.933, *p* < 0.01]. No differences in AP were observed between the PO and CTRL groups from 4 [(F(2,12) = 1.106, *p* = 0.91811] to 25 Hz [(F(2,12) = 0.412, *p* = 0.69128] (Figure 9).

## 4. Discussion

To the best of our knowledge, this is the first study to evaluate the effect of long-term supplementation of FO on the behavioral parameters of adult male rats with early-life history of exposure to hyperthermia, as well as on EEG activity and SWDs from the amygdala during sudden arrest behavior. We hypothesized that long-term intake of FO would provide considerable neuroprotection against the behavioral parameters initially present on pups under early-life HP exposure and consequent in adulthood on EEG hyperexcitability from the amygdala. This hypothesis was confirmed by the behavioral and EEG results, where an attenuation in the severity of the behaviors under HP and a decreased EEG hyperexcitability of the amygdala on SWD without seizure manifestation was observed.

Considering that the embryo is dependent on the maternal supply of θ-3 fatty acids because it cannot produce its own [46], there is a direct relationship between the supply of θ-3 during pregnancy and lactation and better embryo development during the fetal and newborn stages and into adulthood [47]. In the current study, the progenitor dams were supplemented with FO from puberty through mating, delivery, and lactation. Once born, their pups were exposed to 30 min of HP in a glass chamber to gradually raise their CT with hot airflow. 

Several studies have used this HP model to study their effects on the incidence of FS in pup rats. However, the HP treatment in those studies was applied after ten days of age [23,24] with the goal of finding a coincidence with the same age in human beings. In this work, we considered the fact that reduced activity of the GABAergic system is involved in the pathogenesis of HP-induced seizures in immature rats [48]. GABA is the principal inhibitory neurotransmitter of the CNS, and as was mentioned, it is synthesized through the decarboxylation of glutamate by means of the glutamate decarboxylase (GAD), which possesses high heat thermolability into the first five days post-uterine life, but not after six days [25,26]; hence, it is likely that the incipient GABAergic system at five days of age explains the high elevation of CT in the PO and CTRL groups.

The pups exposed to HP presented a gradual increase in their CT during this experiment that likely resulted from changes in internal physiological parameters, such as the activation of proinflammatory mediators [49,50], over-regulation of hypothalamic structures involved in CT regulation under seizure conditions [51,52], or activation of opioid receptors that play a role in regulating body temperature [53].

Opioid receptors of the periaqueductal gray matter are involved in activating (Mu) or inhibiting (Kappa) thermogenesis. Thus, they play a crucial role in many physiological processes as possible mediators of thermal responses caused by stress, among other effects [54]. The activation of Mu receptors in the CNS has a net excitatory, proconvulsant effect [55,56], which is often accompanied by behavioral manifestations of convulsive activity, such as myoclonic twitches or stereotypies. Hence, it is likely that in the pups treated with FO, inhibition of the up-regulation of body temperature could occur as a result of the θ-3 PUFAs molecules contained in FO, probably antagonizing Mu receptors and attenuating the elevation of CT. This speculation may be supported by other studies where, for example, supplementation of θ-3 for eight weeks in a model induced-morphine was associated with reduced anxiety and stress during seeking behavior, and the effect on inhibition of Mu receptor by θ-3 was proposed [57]. Also, the fact that low doses of naloxone (an opioid receptor antagonist) in naïve rats with repeated FS decreased susceptibility to seizures in the mature period with reduced neural damage [58] supports this speculation.

During 30 min of early HP, intermittent walks, sudden arrest, uncontrolled postures, falls with hind limb extension, and subsequent myoclonus events are considered representative of FS [41,52]. It has been reported that the latency for the occurrence of FS under exposure to HP was 10.8 ± 0.6 min [49]. We found a similar result in our work, but only in the rats of the CTRL and PO groups, while those in the FO group presented a higher latency to presenting FS.

During early-life HP, male rats in the FO group had fewer myoclonic twitches than PO and CTRL, which were accompanied by an increase in the latency to the onset of uncontrolled posture following behavior such as seizures. Thus, it is likely that the DHA and EPA contained in FO could have had indirect effects on early GABA neurotransmission, as indicated by the greater latency to presenting uncontrolled postural movements and the lower frequency and duration of those events, as well as fewer episodes of myoclonus.

Although we did not evaluate early-life EEG during conditions of febrile seizures (because it was necessary to maintain the animals alive into adulthood), it is possible that long-term supplementation with FO from the stage of fetal development could extend neuroprotective effects on the brain development of rats and attenuates damage of neuronal tissue. Similar findings were reported recently with supplementation of DHA during the pregnancy of progenitor mice and the infancy of their offspring, with increased levels of DHA in the cerebral cortex associated with a decrease in susceptibility to febrile seizures in offspring. The authors showed the efficacy of DHA on febrile seizures and could provide a new countermeasure against epilepsy after growth in terms of the prevention of febrile seizures [34,35].

Our data showed that the adult rats in all three groups with a history of early HP displayed abnormalities in their EEG from the right and left amygdala that were consistent with the presence of SWDs. These events have been recorded in animal models of absence epilepsy with spontaneous pauses in behavior (sudden arrest), specifically when rats adopt a frozen posture with subtle, spontaneous chewing and head-bobbing movements [16,17,18,19]. In the present study, the rats of the FO group presented a lower number of SWDs than the PO and CTRL groups, which presented a greater frequency and duration of SWDs with a higher number of total SWD events, while FO rats showed a greater latency to and lower frequency of SWDs. This data shows a lower susceptibility to the adverse effects of early-life HP and could result from a minor alteration of the neural activity in cerebral areas involved in the generation of the FS. Alterations of the components and physiology of neurons have been observed as long-term consequences of early FS [59,60]; hence, a possible explanation of our findings is that adult rats with a history of early HP exposure develop a persistent modification of neuronal function with a proclivity toward overexcitability in limbic structures like the amygdala. However, long-term FO supplementation could attenuate SWDs.

The spectral analysis before the occurrence of SWDs showed that it was only in the FO group that the EEG recordings from the left amygdala presented a lower AP from 4–12 Hz compared to the PO and CTRL groups. A similar pattern was obtained after the occurrence of the SWDs in the FO group. The fact that mainly PO and CTRL groups presented a higher AP of the EEG frequencies is consistent with reports that have documented a higher amplitude of the fast frequencies (similar to SWDs) in patients with prolonged non-epilepticus motor status after FS and in children with primarily corticoreticular seizures that include febrile convulsions [61].

During SWDs, a decreased AP in all EEG spectra was observed in the FO group from 4–25 Hz, compared to the PO and CTRL groups. Similar results were reported under long-term supplementation of FO in adult male rats that presented an attenuated effect on behavioral and amygdaline EEG convulsions induced by applying the convulsant 3-mercaptopropionic [39]. Thus, it is probable that long-term supplementation with FO attenuates neuronal hyperexcitability during SWDs as a result of the long-term effects of DHA and EPA molecules on neural tissue by means of several protective mechanisms, such as blocking sodium channels [62], lipoelectric modification of ion channel voltage gating [63], and blocking of inflammatory chains. Interestingly, estrogen concentration may be an important factor in defining the action of estrogen on convulsions. DHA is a retinoid X receptor (RXR) agonist; therefore, DHA-dependent RXR activation is considered to be involved in increasing estradiol levels of the cerebral cortex in pups, which could reduce the sensitivity to febrile seizures [34], among other beneficial effects of the long-chain fatty acids contained in FO [28,33,57]. Particularly, DHA can exert its effects by modifying the physical organization of lipid rafts in neuronal membranes. It has been reported that DHA increases the size and stability of lipid rafts, which affects subsequent signal transduction [64].

It is likely that the activation of Na^+^ currents was attenuated by FO thanks to its DHA and EPA content. The spike burst component of SWDs has been related to persistent Na+ currents that could contribute to depolarization, while wave components have been related to the activation of K^+^ currents [65,66]. Similarly, studies have reported the protective effects of θ-3 molecules against overactivation of K^+^ and Na^+^ channels on neuronal alterations [67,68]. In the present study, SWDs were observed in all three groups, but only FO presented a lower AP during those episodes. Therefore, it is probable that the protective effect of long-term FO supplementation from fetal development onward could be related to the lower AP of the SWDs of adult rats with an early history of HP. Further studies are necessary to explore the effect of the long-term supplementation of FO in animal models of absence seizures. Interestingly, the adult rats with early-life exposure to HP under long-term supplementation of palm oil increased the AP of the SWDs, perhaps due to adverse effects of mechanisms on neural activation. Studies have evidenced the damage of neuronal functions associated with heavy fat consumption in Western diets [69], especially in relation to amygdala activity [70]. Hence, our results support the notion that long-term supplementation of FO may aid in achieving better results in adult rats attenuating damage by neuronal hyperexcitability without seizures.

Palm oil was used in the present study because it contains a different fatty acid profile than fish oil: it contains a higher amount of medium-chain saturated fatty acids and lacks DHA and EPA. Previously, we found that the dose used of fish oil as a supplement during the breeding process (throughout the pregnancy until 5 days old) attenuates febrile seizures induced by hyperthermia in rat pups in comparison with the group supplemented with palm oil [39]. On the other hand, polyunsaturated fatty acids in fish oil are unstable and easily oxidized [71]; therefore, low quantities of natural antioxidants are added to fish oil in order to prevent deterioration.

### Limitations of the Study

The present study has some limitations, principally the fact that the number of male rats used was minimal due to the complex procedures required to maintain them into adult age, especially after the procedure to implant electrodes in their brains.

To complement our work, we recommend obtaining samples of brain tissue to determine, in detail, the cytoarchitecture of the amygdala and ascertain the levels of fatty acids in the different experimental groups.

## 5. Conclusions

Long-term supplementation with fish oil in newborn male rats increases the latency and decreases the frequency and duration of behaviors characteristic of febrile seizures induced by hyperthermia.

Long-term supplementation with fish oil attenuates amygdala hyperexcitability in adult male rats with an early-life history of febrile seizures during arrest behavior but without manifestation of seizures.

The HP model of the rat pups at five days of age could be suitable for the study of absence epilepsy in young and adulthood.

Further studies are necessary to test whether the neuroprotective effect of fish oil extends to adult animals with a history of early febrile seizures when they are provoked by a convulsant.

## Figures and Tables

**Figure 1 brainsci-15-00395-f001:**
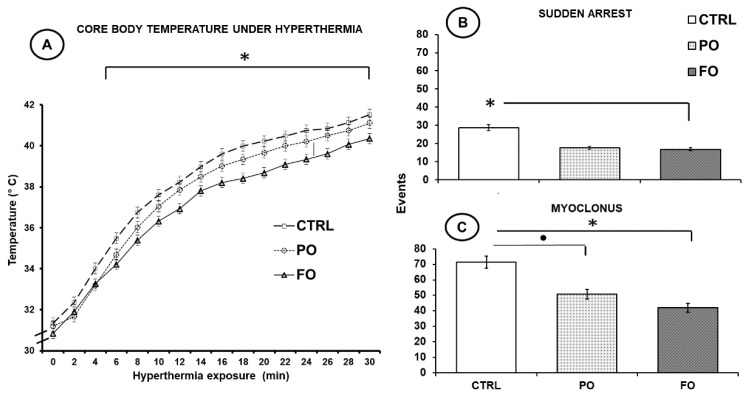
Mean ± SE of core temperature (**A**), sudden arrest (**B**), and myoclonus (**C**) in pups at 5 days of age during exposure to HP. One-way ANOVA followed by a Tukey’s test. * *p* < 0.05 FO compared to PO and CTRL; • *p* < 0.05 CTRL compared to PO and FO.

**Figure 2 brainsci-15-00395-f002:**
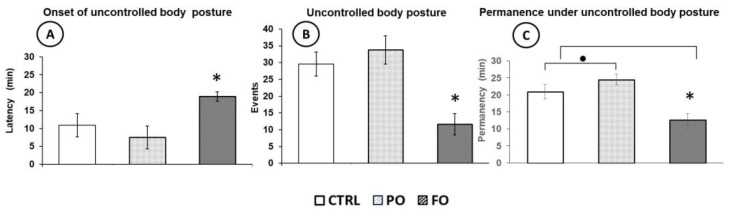
Mean ± SE of the latency to onset of uncontrolled body posture (**A**), frequency of events on uncontrolled body posture (**B**), and permanence in uncontrolled posture during exposure to hyperthermia (**C**). One-way ANOVA followed by a Tukey’s test. * *p* < 0.05 FO compared to PO and CTRL; • *p* < 0.05 PO compared to CTRL and FO.

**Figure 3 brainsci-15-00395-f003:**
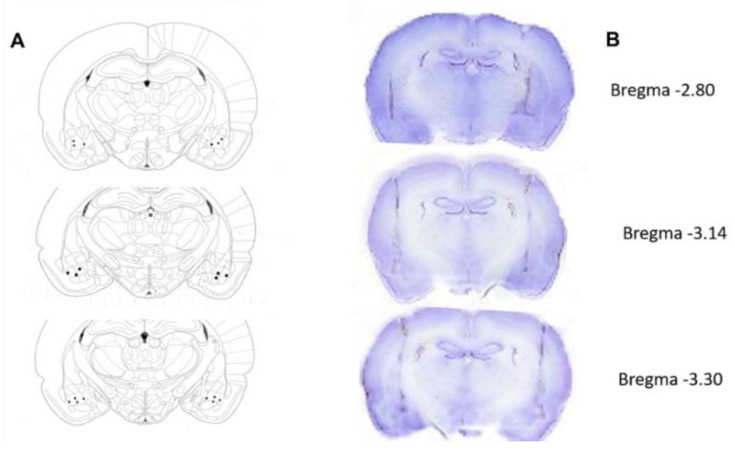
(**A**) Depiction of the sites of the electrode tips implanted in the left and right amygdala of adult rats with an early-life history of HP according to the coronal sections as per the atlas of Paxinos and Watson [42]. (**B**) Three examples of coronal sections of the rats’ brains show the trajectory of the electrodes. The dots show the location of the electrode tips.

**Figure 4 brainsci-15-00395-f004:**
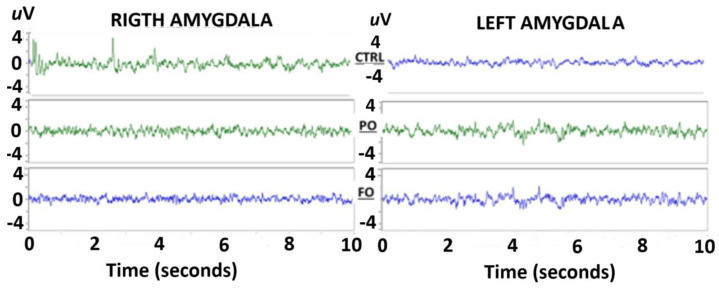
Traces of the EEG of the rats with no early-life history of HP (NT). No abnormalities, such as SWDs, were observed during sudden arrest behaviors in either the right or left amygdala.

**Figure 5 brainsci-15-00395-f005:**
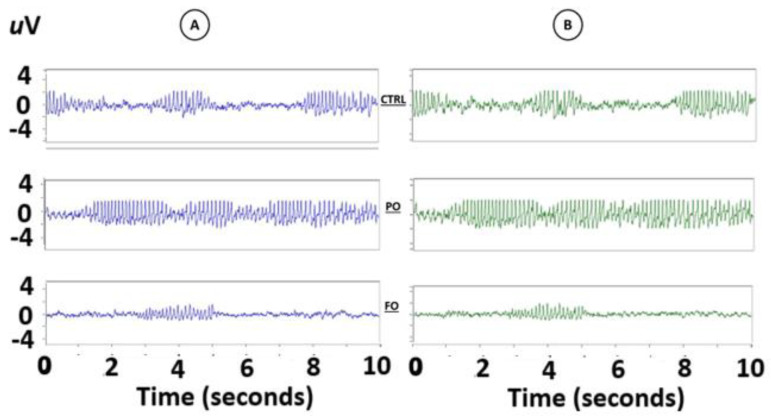
EEG traces from adult rats with an early-life history of HP in the arrest behavior condition from the right (**A**) and left amygdala (**B**). Note the occurrence of SWDs in the PO and CTRL groups: interestingly, synchronized electrical activity between the two brain hemispheres was also observed.

**Figure 6 brainsci-15-00395-f006:**
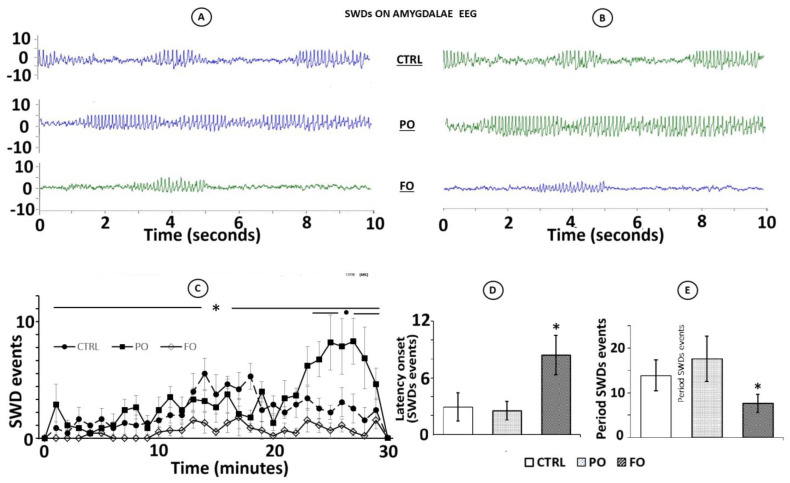
EEG segments from the right (**A**) and left (**B**) amygdala of adult rats with early-life HP correspond to the CTRL (*n* = 7), PO (*n* = 8), and FO (*n* = 5) groups. Mean ± SE of the period of appearance (**C**), onset (**D**), and total number of the spike-wave discharges (SWD) (**E**). One-way ANOVA followed by a Tukey’s test * *p* < 0.05 FO compared to PO and CTRL • *p* < 0.05 PO compared to CTRL and FO.

**Figure 7 brainsci-15-00395-f007:**
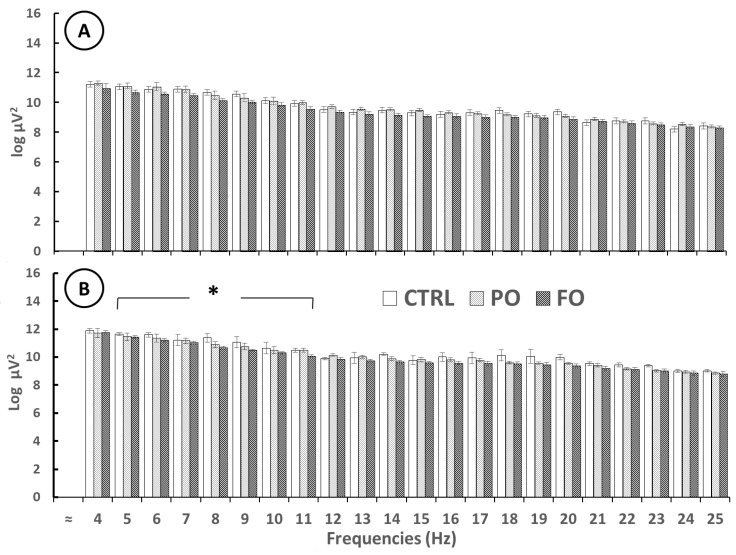
Absolute power (log, mean ± SE) of the several EEG frequencies before SWDs in the adult rats with early-life HP in the right (**A**) and left (**B**) amygdala of the CTRL, palm oil (PO), and fish oil (FO) groups. One-way ANOVA followed by a Tukey’s test. * *p* < 0.05 FO compared to CTRL and PO at each Hz analyzed.

**Figure 8 brainsci-15-00395-f008:**
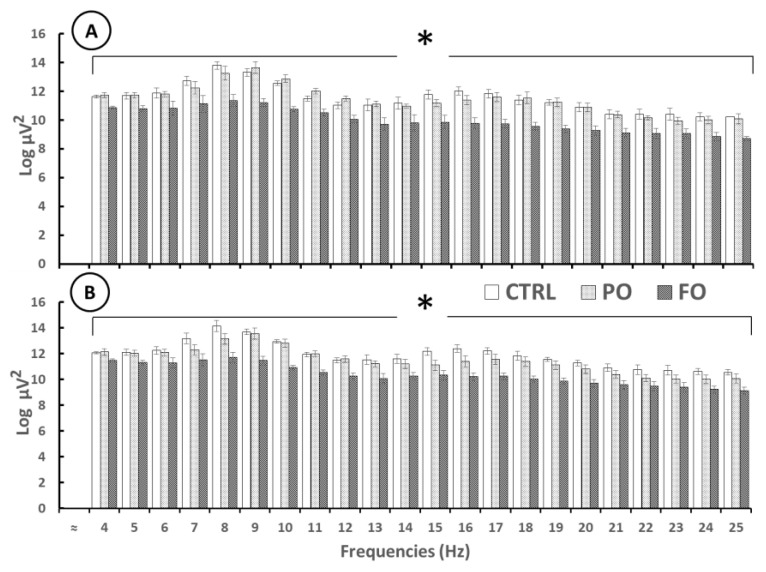
Absolute power (log, mean ± SE) of several EEG frequencies during SWD events in the adult rats with early-life HP in the right (**A**) and left (**B**) amygdala of the CTRL, PO, and FO groups. One-way ANOVA followed by a Tukey’s test. * *p* < 0.05 FO compared to CTRL and PO at each Hz analyzed.

**Figure 9 brainsci-15-00395-f009:**
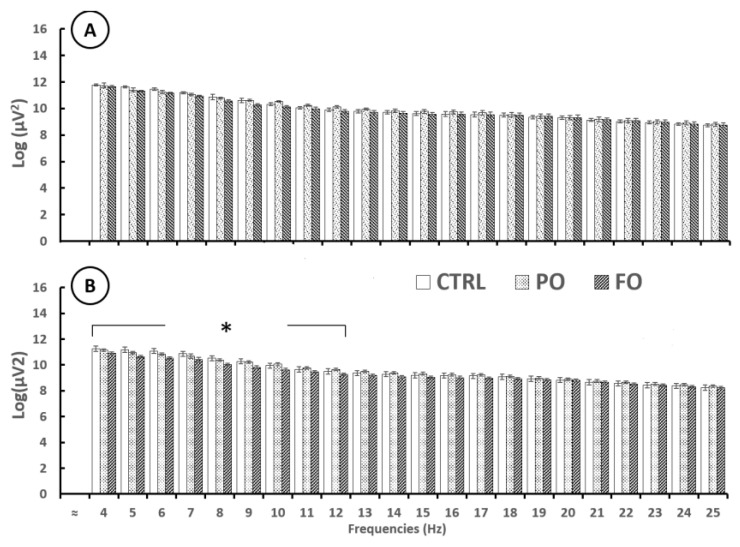
Absolute power (log, mean ± SE) of the EEG frequencies after the occurrence of SWDs in the adult animals with an early-life history of HP in the right (**A**) and left (**B**) amygdala of the CTRL, palm oil (PO), and fish oil (FO) groups. One-way ANOVA followed by a Tukey’s test. * *p* < 0.05 FO compared to CTRL and PO at each Hz analyzed.

## Data Availability

The raw data that supports the findings of the study are already included in the article. Any additional questions can be directed to the corresponding author.

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
