# Peer review of "Long-Term Fish Oil Supplementation Attenuates Spike Wave Discharges in the Amygdala of Adult Rats with Early-Life Febrile Seizures"

_brainsci, 2025, doi:10.3390/brainsci15040395_

Round 1
Reviewer 1 Report
Comments and Suggestions for Authors
This manuscript is clear, well-reasoned, and well-structured.
Cited publications have only 5 out of 70 works in the last 5 years (since 2020), and many citations earlier than 25 years from the date of manuscript preparation (earlier than 2000). It is recommended to add more contemporary studies (within the last 5-7 years) or replace older references to justify the actuality of the research topic.
The manuscript contains a scientifically valid experiment design. However, further justification is required for the choice of fish oil dosage for the animal experiment. Justification is also needed for the choice of palm oil to be used in the experiment in the comparison group with other groups.
The results in the manuscript are stated in a form that allows reproduction of the stages of the experiment.
The figures presented are informative and available for interpretation. Statistical analyses have been applied correctly.
All statements and conclusions are coherent and supported by the citations provided.
The ethical statements are adequate.
I would like to point out that my review contains the following criticisms and queries:
- Cited publications have only 5 out of 70 works in the last 5 years (since 2020), and many citations earlier than 25 years from the date of manuscript preparation (earlier than 2000). It is recommended to add more contemporary studies (within the last 5-7 years) or replace older references to justify the actuality of the research topic.
- The choice of fish oil dose for the animal experiment must be justified.
- The choice of palm oil to be used in the experiment must also be justified.
More than this, I have no critical comments to make on this manuscript.
Author Response
Comment: Cited publications have only 5 out of 70 works in the last 5 years (since 2020), and many citations earlier than 25 years from the date of manuscript preparation (earlier than 2000). It is recommended to add more contemporary studies (within the last 5-7 years) or replace older references to justify the actuality of the research topic.
Response: We update bibliographic references as far as possible.
Comment: The manuscript contains a scientifically valid experiment design. However, further justification is required for the choice of fish oil dosage for the animal experiment. Justification is also needed for the choice of palm oil to be used in the experiment in the comparison group with other groups.
Response: At the last paragraph of the discussion section we indicated that palm oil was employed in control group as fatty substance but without DHA or EPA.
Comment: The results in the manuscript are stated in a form that allows reproduction of the stages of the experiment. The figures presented are informative and available for interpretation. Statistical analyses have been applied correctly. All statements and conclusions are coherent and supported by the citations provided. The ethical statements are adequate.I would like to point out that my review contains the following criticisms and queries: Cited publications have only 5 out of 70 works in the last 5 years (since 2020), and many citations earlier than 25 years from the date of manuscript preparation (earlier than 2000). It is recommended to add more contemporary studies (within the last 5-7 years) or replace older references to justify the actuality of the research topic.
Response: We update bibliographic references as far as possible.
Comment: The choice of fish oil dose for the animal experiment must be justified. The choice of palm oil to be used in the experiment must also be justified.
Response: We added the following text at the end of the discussion: Palm oil was used in the present study because it contains a different fatty acid pro-file than fish oil: it contains a higher amount of medium-chain saturated fatty acids and lacks DHA and EPA. Previously, we found that the dose used of fish oil as a supplement during the breeding process (throughout the pregnancy until 5-days old) attenuates febrile seizures induced by hyperthermia in rat pups in comparison with the group sup-plemented with palm oil [39].
The doses of fish oil and palm oil used are those we previously applied as a supplement in a previous study. We found that chronic fish oil supplementation during the breeding process (throughout the pregnancy until 5-days old) attenuates febrile seizures induced by hyperthermia in rat pups.
Reviewer 2 Report
Comments and Suggestions for Authors
This manuscript researched the effects of fish oil supplementation on the rats with early-life febrile seizures, showing beneficial efficacy and also decreasing the occurrence and amplitude of spike wave discharges. Overall, this study is interesting and may provide indicative hope for managing febrile seizures. However, I have some concerns needing to be addressed before it can be accepted for publication. I also suggest that authors go through the whole manuscript to carefully check all sentences and words.
Major comments.
- For Abstract part, current version is too tedious, and it can be improved with more concise and clear writing.
- For Introduction part, authors cited many references and presented long background. However, going through current introduction makes me feel not so organized. I recommend that authors may revise to trim unrelated description.
- For Results part, such as 3.2, 3.4, 3.5, and 3.6, authors made their conclusions as main text in these studies, but there are no data presented for supporting them. Why do not show these data if already tested by authors? There are too many “data not shown” in these parts, making readers feel insufficiency of evidence.
Minor comments.
- Line 57, “for example:” to “for example, “
- Line 63, are to is
- Line 84, catabolize to catabolizes
- Line 100, contain to containing
- Line 109, delete it
- Line 111, have to has
- Line 363 and line 375 have the same subtitle “3.9”
- Line 388, add it before “is”
English expression should be improved.
Author Response
Comment: For Abstract part, current version is too tedious, and it can be improved with more concise and clear writing.
Response: The abstract was modified as suggested
Comment: For Introduction part, authors cited many references and presented long background. However, going through current introduction makes me feel not so organized. I recommend that authors may revise to trim unrelated description.
Response: The introduction was modified and we deleted unnecessary information.
Comment: For Results part, such as 3.2, 3.4, 3.5, and 3.6, authors made their conclusions as main text in these studies, but there are no data presented for supporting them. Why do not show these data if already tested by authors? There are too many “data not shown” in these parts, making readers feel insufficiency of evidence.
Response: Data where no significant differences are indicated may be included as supplementary information
Comment: Minor comments.
Line 57, “for example:” to “for example, “
Line 63, are to is
Line 84, catabolize to catabolizes
Line 100, contain to containing
Line 109, delete it
Line 111, have to has
Line 363 and line 375 have the same subtitle “3.9”
Line 388, add it before “is”
Response: the text was now corrected
Comments on the Quality of English Language: English expression should be improved.
Response: The grammar was revised and corrected.
Reviewer 3 Report
Comments and Suggestions for Authors
The article entitled "Long-term fish oil supplementation attenuates spike wave discharges in the amygdalae of adult rats with early-life febrile seizures” investigates the effect of long-term FO supplementation on the severity of febrile seizures”. It is a well-structured article with a clear research question and supporting experiments, but there are some points that should be considered to improve the final quality as follows:
1. The term amygdalae is written many times within the manuscript and sometimes written as amygdala, amygdalae is a Greek word. The English word is amygdala. Please, correct it through the whole manuscript.
2-Materials and Methods: Fish oil that was used in the experiment was mixed with tocopherols, ascorbyl palmitate, and citric acid. How did the authors avoid the activity of other ingredients within the FO. In other words, how did they confirm there is no cross-activity between these ingredients?
3. Histological verification: Figure 3 needs to be coloured to show the differences between the coronal sections of the rats’ brains.
4-Discussion: Authors clearly explained the expected mode of action of FO on amygdala activity based on biological features, but they didn’t explain the mode of action based on the structural features of FO. Write a short paragraph that describes structural features of FO and how they induced this activity.
5-Conclusion: Authors need to reveal the importance of their study by explaining the expected outcomes and future perspective of the study.
6-References: The authors used old references like No. 55, which is very old (1984). Kindly update the old references that are below 2000 with updated, well-established references.
7. There are some linguistic and grammatical mistakes, like:
Line 109: The Amy it is often involved = the Amy is often involved
Line 418: Opioid receptors that plays = opioid receptors that play
Carefully review the whole manuscript and correct any type of mistakes.
Comments on the Quality of English LanguageThe English language has a few mistakes that need to be revised.
Author Response
Comment: The term amygdalae is written many times within the manuscript and sometimes written as amygdala, amygdalae is a Greek word. The English word is amygdala. Please, correct it through the whole manuscript.
Response: The text was corrected as suggested
2-Materials and Methods: Fish oil that was used in the experiment was mixed with tocopherols, ascorbyl palmitate, and citric acid. How did the authors avoid the activity of other ingredients within the FO. In other words, how did they confirm there is no cross-activity between these ingredients?
Response: We added the following text at the end of the discussion: On the other hand, polyunsaturated fatty acids of fish oil are unstable and easily oxidized [70], therefore low quantities of natural antioxidants are added to fish oil in order to prevent deterioration.
Comment: Histological verification: Figure 3 needs to be coloured to show the differences between the coronal sections of the rats’ brains.
Response: We feel that the black and white image is clear.
Comment: Authors clearly explained the expected mode of action of FO on amygdala activity based on biological features, but they didn’t explain the mode of action based on the structural features of FO. Write a short paragraph that describes structural features of FO and how they induced this activity.
Response: The text was modified in the discussion: DHA is incorporated into the plasma neuronal membranes to maintain the function of cells by controlling membrane fluidity.
5-Conclusion: Authors need to reveal the importance of their study by explaining the expected outcomes and future perspective of the study.
Response: Conclusion was slightly modified
Comment: The authors used old references like No. 55, which is very old (1984). Kindly update the old references that are below 2000 with updated, well-established references.
Response: References were updated as far as possible.
Comment: There are some linguistic and grammatical mistakes, like:
Line 109: The Amy it is often involved = the Amy is often involved
Line 418: Opioid receptors that plays = opioid receptors that play
Response: grammatical mistakes were corrected
Comment: Carefully review the whole manuscript and correct any type of mistakes.
Response: The manuscript was extensively revised
Comments on the Quality of English Language
The English language has a few mistakes that need to be revised.
Response: the whole manuscript was revised as suggested
Round 2
Reviewer 2 Report
Comments and Suggestions for Authors
The manuscript is improved much, recommending it can be accepted for publication. Congratulations!